# Experimental, Theoretical and Numerical Studies on Thermal Properties of Lightweight 3D Printed Graphene-Based Discs with Designed Ad Hoc Air Cavities

**DOI:** 10.3390/nano13121863

**Published:** 2023-06-15

**Authors:** Giovanni Spinelli, Rosella Guarini, Rumiana Kotsilkova, Evgeni Ivanov, Vittorio Romano

**Affiliations:** 1Faculty of Transport Sciences and Technologies, University of Study “Giustino Fortunato”, Via Raffaele Delcogliano 12, 82100 Benevento, Italy; 2Institute of Mechanics, Bulgarian Academy of Sciences, Acadamy. G. Bonchev Str., Block 4, 1113 Sofia, Bulgaria; rgrosagi@gmail.com (R.G.); kotsilkova@yahoo.com (R.K.); ivanov_evgeni@yahoo.com (E.I.); 3Research and Development of Nanomaterials and Nanotechnologies (NanoTech Lab Ltd.), Acad. G. Bonchev Str., Block 4, 1113 Sofia, Bulgaria; 4Department of Industrial Engineering, University of Salerno, Via Giovanni Paolo II, 84084 Fisciano, Italy; vromano@unisa.it

**Keywords:** PLA-based filament, 3D printing (FDM), thermal conductivity, numerical simulation

## Abstract

The current state of the art on material science emphasizes recent research efforts aimed at designing novel materials characterized by low-density and advanced properties. The present article reports the experimental, theoretical and simulation results on the thermal behavior of 3D printed discs. Filaments of pure poly (lactic acid) PLA and filled with 6 wt% of graphene nanoplatelets (GNPs) are used as feedstocks. Experiments indicate that the introduction of graphene enhances the thermal properties of the resulting materials since the conductivity passes from the value of 0.167 [W/mK] for unfilled PLA to 0.335 [W/mK] for reinforced PLA, which corresponds to a significantly improvement of 101%. Exploiting the potential of 3D printing, different air cavities have been intentionally designed to develop new lightweight and more cost-effective materials without compromising their thermal performances. Furthermore, some cavities are equal in volume but different in the geometry; it is necessary to investigate how this last characteristic and its possible orientations affect the overall thermal behavior compared to that of an air-free specimen. The influence of air volume is also investigated. Experimental results are supported by theoretical analysis and simulation studies based on the finite element method. The results aim to be a valuable reference resource in the field of design and optimization of lightweight advanced materials.

## 1. Introduction

Reduction in mass with consequent savings in materials and costs is one of the main factors to be considered in the structural design phase. This is particularly significant in aircraft and spacecraft fields where sandwiched composite structures, often based on honeycomb panels, are increasingly widely used for saving weight while simultaneously ensuring structural performance and mechanical properties [1,2]. Although these light-weight structures are suitable for large loadings, from a thermal point of view, they are not as efficient. Today, design efforts are aimed at achieving novel multifunctional structures with combined remarkable heat transfer features and structural integrity for a variety of applications, including the most recent ones in the electronic industry where new devices operating at ever-increasing frequencies raise critical heat dissipation issues [3].

Polymers are classically recognized as insulating materials, owing to their low electrical and thermal conductivity, and are therefore generally disregarded in similar contexts. However, given their remarkable properties in terms of lightness, cost-effectiveness, easy processing, corrosion resistance and strength-to-weight ratio, polymers are receiving much interest, even in applications of thermal management where heat dissipation is necessary [4,5]. Therefore, it is widely-accepted that developing advanced polymer composites with enhanced thermal conductivity could be an effective solution to meet this requirement. Recent developments in high-performance thermal polymers are based on nanotechnology. The introduction in polymer matrices of highly thermally conductive fillers such as the carbon-based ones seems effective to improve the thermal conductivity of resulting materials [6,7,8]. This because the consequent creation of suitable conductive paths favors the mechanism of heat conductivity, which is driven by the ability of materials to conduct phonons [9]. Thermal conductivity is governed by the concentration of heat conductive filler dispersed, its intrinsic thermal properties, functionalization and aspect ratio, the thermal conductivity of the host matrix and any interfacial gaps which represent the most crucial barrier for heat transfer [10]. Although different thermally conductive fillers such as ceramic [11,12] and metallic [13,14] particles have been successfully adopted, carbon-based fillers have attracted greater attention due to their shapes and high aspect ratio, which are favorable parameters to easily form thermally conductive network at lower concentrations without degrading the mechanical properties of the resulting composites [15,16]. For instance, in more recent times, graphene and its derivates are considered the most promising fillers due to their good dispersion and better phonon transport at the interface with a wide range of polymeric matrices [17,18]. However, in the evolution of high-performance graphene-based nanocomposites, to date, the improvement of their thermal conductivity through nanotechnology is still limited and the achieved values do not approach those of traditional metals. Functionalization of the filler often tends to enhance the structural morphology in order to reduce phonon scattering and interfacial thermal resistance, thus improving the thermal properties [19]. Thus, in light of this, Yadav and Cho presented an effective method to design high-performance polyurethane (PU) nanocomposites with improved mechanical and thermal features via the introduction of functionalized graphene nanoplatelets during in situ polymerization [20]. A significant improvement in mechanical and thermal properties is also observed by Shukla and Sharma in epoxy hybrid composites using functionalized graphene and carbon-nanotubes [21], and in functionalized-graphene/ethylene vinyl acetate co-polymer composites by Kuila et al. [22]. Moreover, the overall physical characteristics of the nanocomposites, such as electrical and thermal conductivity, can be also related to their fabrication processes. In recent years, the new and innovative additive manufacturing (AM) alias 3D printing has emerged and attracted interest for the great opportunity it provides to convert virtual 3D models created via a computer-aided design (CAD) into physical objects, without the need for molds or machining [23,24,25]. As a result of this versatility, AM is becoming one of the major technologies in the fabrication of fully customized objects that can be realized in complex shapes that could be more effective in thermal transfer applications, such as polymer-based heat sinks.

Pezzana et al. have proposed a 3D printed silicone acrylate-based formulation (Polydimethylsiloxane, PDMS) with improved thermal conductivity as a result of the introduction of boron nitride (BN) [26]. Ji et al. have presented 3D printed composites including alumina and carbon nanofibers (CFs) that show enhanced thermal properties with respect to those exhibited by cast composites [27]. The influence of different metal particles dispersed in PLA-based nanocomposites, prepared via 3D printing with different printing parameters, on the thermal conductivity of the resulting structures has been analyzed in Laureto et al.’s work [28].

Timbs et al. have presented 3D printed heat sinks based on thermally conductive polymer composites with oblique fins, which present a lower thermal resistance and a better convective heat transfer, compared to the straight finned typical of heat exchangers [29,30]. For the purpose of improving thermal management features of GaN transistors, Gerges et al. have proposed a light and cheap 3D printed polymer-based heat dissipator [31].

In our previous study, the thermal behavior of poly (lactic acid) PLA composites reinforced with several carbon-based fillers produced via melt compounding have been experimentally and numerically investigated for their potential use in thermal applications such as heat sinks [32]. Yet, in another paper of ours, the effect of temperature on the thermophysical properties of 3D printed PLA-based composites reinforced with two different type of graphene nanoplatelets (GNPs) was taken into account with the promise that in a future paper more particular and more complex discs designed with AM technology would be considered [33]. The main goal and novelty of the present paper is to compute—using a hybrid experimental/theoretical and computational approach—the effective thermal conductivity of 3D printed discs, based on PLA and 6 wt% of GNPs, with designed ad hoc air cavities, different in the geometry and volume. It is important to underline that these complex samples were produced due to the extraordinary versatility of design offered by 3D printing technology; with traditional manufacturing processes, it would have been quite difficult. As better described in the next section, Materials and Methods, the experimental setup involves an accurate laser flash apparatus, and the theoretical support refers to thermal circuit theory, whereas the simulation tool is based on the finite element method.

The reported results aim to be a general guideline for researchers involved in the design and production of multifunctional and advanced nanocomposites with lightness features and negligible degradation of the thermal properties.

## 2. Materials and Methods

Ingeo™ Biopolymer PLA-3D850 (Nature Works, Minnetonka, MN, USA) is specifically developed for producing a 3D printing monofilament in order to achieve excellent processability and printability, as well as enhanced impact and heat resistance of the printed parts; as such, it is used in the present study for the experimental activity. Moreover, filaments based on Ingeo PLA show remarkable 3D printing features such as less warping or curling and hence precise details, good adhesion to building plates (also in absence of heating) and are odorless while printing. In order to obtain a nanocomposite filament with improved thermal properties, the above polymer matrix has been enriched with graphene nanoplates (TNIGNP, from Times Nano, Chengdu, China). The main physical properties of the host polymer, filler and resulting nanocomposite filaments are briefly illustrated in the schematic representation of Figure 1. For the sake of completeness, it is worth pointing out that X-ray diffraction (XRD), differential scanning calorimetry (DSC) and Raman spectroscopy analyses of the GNP nanosheets have been carried out in our previous studies [34,35]. The results show a high crystalline quality and a multilayer structure of the GNPs with a number of layers < 30.

Nanocomposite compounds have been produced by using a melt extrusion technique. Both PLA pellets and GNPs were blended by maintaining the temperatures in the range 170–180 °C, in a twin screw extruder (COLLIN Teach-Line ZK25T, Maitenbeth, Germany) operating with a screw speed of 40 rpm. As a masterbatch, nanocomposites at 12 wt.% of loadings were obtained, whereas, in a second extrusion run, a formulation based on 6 wt.% of total filler content was obtained by diluting the masterbatch with pure PLA in the right amount. Pellets of the compound analyzed in the current work are the result of two extrusion runs. The concentration of 6 wt% has been selected based on the experimental results observed in our previous studies [33,36]. In brief, this specific filler concentration ensures simultaneously interesting electrical, mechanical and thermal properties combined with suitable rheological features in terms of viscosity for realizing high-quality 3D printed parts needed for performing the analysis planned in the present work.

The FDM filaments based on pure PLA or 6 wt% of GNPs with a diameter of 1.75 mm were obtained using single screw extruder (Friend Machinery Co., Zhangjiagang, China) on the aforementioned nanocomposite pellets. A speed of 20 rpm and a temperature range of 170–220 °C was set for this purpose. Finally, a quenching in water bath at 60 °C accomplished the required gradual cooling. Disc-shape specimens with a diameter of 12.6 mm and a thickness of 3 mm were 3D printed with a fused deposition modeling (FDM) process through a German RepRap 3D printer X-400 Pro (German RepRap GmbH, Feldkirchen, Germany). Three samples for each designed configuration were printed. The production line from the beginning of the filament extrusion up to 3D printed samples is schematically illustrated in Appendix A, whereas the main selected printing parameters are reported in Appendix A.

Scanning electron microscopy (SEM) analysis was carried out to obtain information about the interaction between matrix material and nanofiller, as well as its dispersion state. For this aim, a field emission SEM apparatus JSM-6700F (JSM-6700F, Jeol, Akishima, Japan) was used with suitable specimens prepared with a procedure based on the following steps, previously described in Spinelli et al.’s work [37,38]: (i) ad hoc fractured in liquid nitrogen; (ii) chemical etching to remove excess material; (iii) an ultra-thin coating of electrically conducting metal (gold AU, given its high conductivity and relatively small grain size) to improve the imaging of samples due to the increases of the signal to noise ratio.

The thermophysical properties (thermal conductivity, thermal diffusivity and heat-capacity) of the test samples have been investigated by means of laser flash analysis (LFA) which has proven itself as a fast, versatile and precise measuring method for a variety of different materials, since its first introduction by Parker et al. in 1961 [39]. LFA is based on the physical principle previously reported in Spinelli et al. [33] and here, meticulously described and graphically represented (Appendix A) in the Appendix A.

In the present study, disc-shaped samples, with a diameter of 12.6 mm and a thickness of 3 mm and different air cavities, were tested with the Laser Flash Technique (LFA 467 Hyper flash, Neztsch, Germany) at room temperatures of 293.15 K. As schematically illustrated in Table 1 some air cavities with different geometry but equal volumes have been realized in order to evaluate the influence of the shape on the thermal properties of the resulting samples. Table 2 summarizes instead the samples used to investigate the influence of the progressive increase in the total volume of air obtained by increasing the number of appropriate cavities. Finally, Table 3 shows the samples with air cavities based on a truncated cone (with equal volumes) designed to study the influence of their radiuses and their orientations. All the geometric details and the formalism used for their design have been reported, from time to time, in each table.

The samples were measured three times at the indicated temperature, whereas the results reported in the relative section refer to the average values. In accordance with the guidelines supplied with the device, prior to the measurements, both the faces of the samples were coated with graphite spray to improve the emission/absorption features of the samples. The specific heat was calculated using the reference method according to the standardization ASTM-E 1461–2011. Consequently, the LFA was calibrated with a Cp-standard (Pyroceram square sample with 10 mm in size and a thickness of 2 mm). The density of the 3D printed specimens was determined by means of buoyancy flotation method at room temperature.

In the present work, the thermal behavior of the designed nanocomposites with different cavities filled by air was numerically analyzed by means of an accurate simulation study based on a 3D finite element method (FEM).

The commercial software COMSOL Multiphysics^®^ (version 5.5) has been adopted as a simulation environment in which the experimentally tested specimens have been faithfully reproduced in all details: each CAD (computer-aided design) is exactly the same as that used for 3D printing. 

A schematic representation of the addressed case study together with the most important model definitions is reported in Figure 2a and Figure 2b, respectively.

The thermophysical properties of the simulated samples are evaluated starting from an initial room temperature (293.15 K) in adiabatic condition because this mode enables a better determination of the influence of the air cavities.

A rectangular pulse of intensity of 1200 W/m^2^ and duration of 60 s has been applied to the undersurface of the sample to heat it.

The heat transfers inside the body from the heated surface to the opposite face via conduction, according to the Fourier’s equation and the thermal energy equation, as detailed described in the Appendix A.

Here, in short, it is worth mentioning that the thermal energy equation (in Cartesian coordinate system) to describe the conductive transport at constant pressure on a differential volume Δx∙Δy∙Δz, can be written as follows [40]:

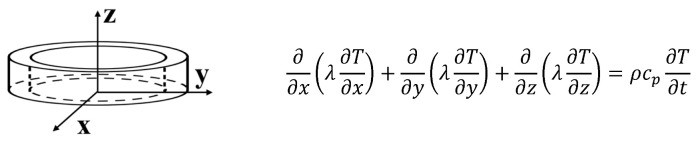
(1)
where:
ρ is the density of the material [kg/m^3^];*c_p_* is the specific heat of material [J/kg∙K];λ is the thermal conductivity of the involved material, [Wm∙K];$\lambda \frac{\partial T}{\partial x},\;\lambda \frac{\partial T}{\partial y}$, $\lambda \frac{\partial T}{\partial z}$ are the conductive heat flux components along the x, y and z directions, expressed by Fourier’s law as specified in Appendix A;$\frac{\partial T}{\partial t}$ is the change in temperature over time.

As the initial condition (*t* = 0), the sample is at room temperature T_room_ (293.15 K); as boundary conditions, symmetry on the axis for *x* = 0 and *y* = 0 is assumed, whereas a thermal insulation is considered at the lateral, upper, back and front surfaces; at last, a heat flux is imposed at bottom surface (*z* = 0).

All analytic terms concerning the initial and boundary conditions to solve the thermal energy equation are reported in Table 4.

## 3. Results

### 3.1. Morphological Analysis

The investigation of the morphology of the nanocomposite mixtures in terms of dispersion state and isotropy at microscale level plays a key role in interpreting the overall macroscale behavior and performance of the resulting materials and, in particular, their mechanical, thermal and electrical properties measurements. 

The cryogenically fractured surfaces of test specimens of unfilled PLA (Figure 3a) and nanocomposites containing 6 wt% of GNPs (Figure 3b) were analyzed using scanning electron microscopy. With regard to the pure PLA, as observable in Figure 3a and more clearly in the magnification of Figure 3b, due to the adopted fracturing treatment, the specimens underwent brittle fractures since planar surfaces are created without roughness and evident graininess, which are indicative elements of deformation prior to failure typical of other approaches.

Figure 3b provides an overview of the dispersion state within the matrix of the GNPs particles, which appear in a sort of stacked arrangement. This is because the as-received filler already comes in a stacked form, which is initially maintained during the extrusion process and then further preserved throughout the samples production as a result of the adopted method being based on layer by layer deposition (FDM technology). Such a phenomenon has also been noted with reference to other nanoparticles and layer-by-layer growth mechanisms [41]. This particular morphological arrangement of the GNPs, lowering the interfacial thermal resistance (also known as Kapitza resistance, Rk), ensures improved percolation paths for the thermal transport, as is confirmed by the experimental results.

### 3.2. Thermal Conductivity of Air Cavity with Different Shapes: Experimental, Theoretical and Simulation Results

Figure 4 reports the experimental results regarding the effective thermal conductivity for both full samples and with cavities filled by air in different geometries (i.e., square, circle and triangle), in the case of pure PLA and PLA including 6 wt% of GNPs in (a) and (b), respectively. It is emphasized that the area volumes in this case are the same since the air cavities are equal in height and base area. It is also important to specify that regardless of the shape, the cross-sectional area of the cavity along the thickness remains constant at all heights.

After these necessary introductory statements, it is important to underline how the introduction of graphene particles significantly improves the thermal conductivity of the resulting materials. In the case of samples without cavities, the conductivity goes from 0.167 [W/mK] for pure PLA to 0.335 [W/mK] for PLA filled with 6 wt% of GNPs. This improvement, in percentage terms, corresponds to an interesting 101%. Regardless of the type of composite under consideration, it is worth noting that in the presence of air cavities, the conductivity is reduced of approximately 10%, independently from their geometry. As previously premised in Section Materials and Method, all these air cavities are characterized by a constant cross-section along the z-direction, whereas the air volume is conserved because the base areas and heights are equal. 

An electrical analogy with conduction heat transfer can be useful to explain such experimental results. By assuming the heat rate Q [W] as the analog of the electrical current I [A] and the temperature difference, and Δ*T* [K] as the analog of the voltage difference ΔE [V], it is possible to define the thermal conductance *Kth* [W/K] and the thermal resistance *Rth* [K/W] according to the heat transfer rate given by the following equation:(2)$$Q=Kth\cdot \Delta T=\left(\frac{1}{Rth}\right)\cdot \Delta T$$
where: (3)$$Kth_{\;}=\frac{\lambda_{\;}\cdot A_{\;}}{l_{\;}}=\frac{1}{Rth_{\;}}$$

The thermal resistance and, therefore, the thermal conductance are dependent on the dimensions of the sample considered (length *l* and cross section area *A*, respectively) and its thermal conductivity, *λ*.

Based on the established analogy, any of the nanocomposite systems with one or different air cavities considered in this article, since the heat rate *Q* flows from the heated bottom surface to the top surface of the cylinder in which the air is encapsulated, can be represented by the resistance circuit of Figure 5a. It shows that the heat rate flows in parallel through the materials, with resistances *Rth*2_material_ and *Rth*2_air_, for which the following is true: (4)$$Q=\left(\sum Kth_{i}\right)\cdot \Delta T=\left(\sum \frac{1}{Rth_{i}}\right)\cdot \Delta T$$
where:(5)$$\sum Kth_{i}=Kth2_{material}+Kth2_{air}=Kth2_{p}$$
or
(6)$$\sum \frac{1}{Rth_{i}}\;=\;\frac{1}{Rth2_{material}}+\frac{1}{Rth2_{air}}\;=\;\frac{1}{Rth2_{p}}$$
from which:(7)$$Rth2_{p}=\frac{Rth2_{air}\cdot Rth2_{material}}{Rth2_{air}+Rth2_{material}}$$

By substituting the parameters corresponding to each conductance or resistance in Formulae (5) or (6), the equivalent parallel conductivity $k_{ep}$ is as follows:(8)$$k_{ep}=\lambda_{air}\cdot \left(1-\varphi \right)+\lambda_{material}\cdot \varphi$$
where *φ* is the ratio between the volume of the material and the total volume:(9)$$\varphi =\frac{V_{material}}{V_{material}+V_{air}}$$

On the contrary, the materials with resistance *Rth*1*_material_*, *Rth*2*_p_* and *Rth*3*_material_* are in series (see Figure 5b) with the heat rate *Q*; thus, the following is true:(10)$$Q=\left(\frac{1}{\sum Rth_{i}}\right)*\Delta T=\frac{1}{\sum \frac{1}{Kth_{i}}}*\Delta T$$
where:(11)$$\sum Rth_{i}=Rth1_{material}+Rth2_{P}+Rth3_{material}=Rth_{s}$$
or:(12)$$\sum \frac{1}{Kth_{i}}=\frac{1}{Kth1_{material}}+\frac{1}{Kth2_{p}}+\frac{1}{Kth3_{material}}$$

In the specific case:(13)$$Rth1_{material}=Rth3_{material}=Rth_{material}=\frac{l_{material}}{\lambda_{material}*A_{material}}$$

By substituting the parameters corresponding to each conductance or resistance in Formulae (11) or (12), the equivalent series conductivity $k_{es}$ is as follows:(14)$$k_{es}=\frac{\lambda_{air}*\lambda_{material}}{\lambda_{air}*\varphi +\lambda_{material}*\left(1-\varphi \right)}$$

If the orientation of the component materials is not in parallel or in series as above described, the equivalent thermal conductivity *k_e_* has a value between *k_ep_* and *k_es_* [42]:(15)$$\frac{\lambda_{air}\cdot \lambda_{material}}{\lambda_{air}\cdot \varphi +\lambda_{material}\cdot \left(1-\varphi \right)}=k_{es}\leq k_{e}\leq k_{ep}=\lambda_{air}\cdot \left(1-\varphi \right)+\lambda_{material}\cdot \varphi$$

In our case studies, regardless of the geometry of the air cavity, the overall thermal resistance and, therefore, the overall thermal conductance of the resulting structures is always the same because the parameters of the air bubble are kept constant, as schematically shown in Figure 5c. Consequently, the final thermal conductivity does not vary in the different cases analyzed.

These thermal aspects were also numerically investigated through multiphysics simulation studies, as reported in Figure 6. Furthermore, the temperature profiles (average values) over time (up to 150 s) recorded on the upper, mid and lower surfaces related to the full sample of pure PLA and PLA + 6 wt% GNPs are shown on the left-side of Figure 6a and Figure 6b, respectively.

In line with the experimental results and the theoretical predictions, some fundamental differences are evident between the two investigated nanocomposites due to their different values in the thermal conductivity. Firstly, a clear hysteresis between the two curves relating to the upper and lower surfaces is appreciable in the case of the pure PLA. Differently, with reference to the reinforced composite (PLA + 6 wt% GNPs), the higher thermal conductivity causes a better heat transport in the solid with a consequently smaller thermal difference between the two opposite faces. This different heat propagation is also visible from the 3D graphics relating to the corresponding cross sections in the xy-plane shown in the right side of Figure 6.

Figure 7 shows a temperature multislice representations of this thermal aspect evaluated at time *t* = 60 s. These graphical views allow us to note that the temperature remains constant in each cross section along the entire thickness of the sample due to the absent of air cavities and external heat exchange, which is neglected in this study to better highlight the effect of internal air when it is present (next results).

Since the observed thermal phenomena are physically equal (except for the numerical values) both for samples made with pure PLA and with reinforced PLA, for the sake of brevity, all subsequent simulations will henceforth refer to the most thermally performing material (PLA + 6 wt% GNPs).

Figure 8 shows different views of the thermal profiles inside the samples filled with the aforementioned air cavities with different geometries (square, circle and triangle) but equal volumes. The results previously observed experimentally and interpreted with theoretical knowledge can be also be verified through this multiphysics modelling investigation.

As is clearly visible from graphic representations of Figure 8a–c, respectively, the heat distribution inside the solid (at the time *t* = 60 s) is the same regardless of the geometry of the air cavities.

Such a numerical analysis allows us to notice how the air cavities, which are characterized by a lower thermal conductivity than the material, initially behave as a barrier to the heat flow and then as a trap for it.

A local overheating around and inside the air cavities is quite evident. In addition, Figure 8d shows that, even dynamically, the heat transfer is identical in the three investigated cases since the three curves relating to the evolution of the upper and lower surfaces temperatures over time are perfectly superimposed.

### 3.3. Thermal Conductivity of Nanocomposites with Multiple Air Cavities: Experimental, Theoretical and Simulation Results

Figure 9 shows the experimental results concerning the variation in the thermal conductivity as a function of the number of equal air cavities (N) progressively introduced into the solid in order to investigate the effect of the total air volume on this thermal property.

Additionally, in this case, it is worth to highlight that with these cavities being of a cylindrical type, their cross section area is constant along the entire thickness.

In particular, Figure 9a,b reports the experimental results concerning the pure PLA and PLA + 6 wt% GNPs, respectively.

For both composites, a decrease in thermal conductivity is observed as the number of cavities increases. 

As shown in Figure 9c,d, this decrease is linear since the interpolation of the experimental data is characterized by a coefficient of determination (R^2^) close to 1.

The results agree with a numerical calculation of the thermal conductivity of nanocomposite including a fluid (air) in volumetric proportion φ and 1–φ, respectively [43]. In our case, the nanocomposite has thermal conductivity κ_m_ (0. 167 W/mK for pure PLA and 0.335 W/mK for PLA + 6 wt% GNPs, respectively), whereas for air in normal conditions, a thermal conductivity of *κ_f_* = 0.026 W/mK is assumed.

In our case study (see Figure 10), since the N cylindrical cavities, each of diameter Φ_p_ and volume *v*, are inserted in a disc with diameter Φ_G_ and volume *V* and are traversed in parallel by the heat flow, the effective thermal conductivity $k_{eff}$, evaluated according to Equation (15), can be expressed as follows: (16)$$k_{eff}=\left[\left(V-Nv\right)\cdot k_{m}+Nv\cdot k_{f}\right]/V$$

Predictive expressions of the thermal conductivity are interesting in the first analysis. Figure 11 shows a direct comparison via histogram visual representation of the results obtained experimentally with those theoretically predicted using the formula of Equation (16) for both cases: PLA and PLA + 6 wt% GNPS in (a) and (b), respectively.

It is interesting to note the good agreement between the results given the low relative error (E_r_, in percentage) that is committed with the theoretical estimation. It is noted that Equation (16) negligibly overestimates the experimental conductivity of the structures.

Figure 12 shows a 3D graphic representation of the simulated graphene-based samples that include a different number of cavities (1, 2 and 3 in (a), (b) and (c), respectively) coherently with the cases investigated experimentally.

The choice of this multislice view allows us to better investigate the temperature distribution across the different sections that make up the samples. A conspicuous local overheating is detected in correspondence with each air cavity, whereas the heat diffuses uniformly in the layers in which it meets only material characterized by a high thermal conductivity compared to that of the air. Because of the increasing presence of cavities and therefore of air volume, the overall conductivity of the resulting structure is progressively reduced. This can be deduced from the analysis of the relative color bars. In fact, the temperature excursion scale of the specimen slightly changes in the three cases analyzed. It is closely linked to the thermal conductivity: the greater the latter, the smaller the thermal excursion due to the better heat transfer in the solid, according to Fourier’s thermal law.

Furthermore, with the aid of the simulation software, it was possible to select, as indicated in Figure 13 (left side), a series of cross sections along the thickness of the sample on which to detect the average temperature values at the instant of time t = 60 s in which the thermal source ceases (see Figure 13 right side). Some sections fall exclusively in the material (sections at 0, 0.6, 2.4 and 3 mm) and others in a hybrid material/area condition (sections at 1.2 and 1.8 mm).

Regardless of the number of cavities, the temperature decreases exponentially along the thickness as the quote moves away from the heat source. Obviously, the greater the volume of air compared to the overall volume of the sample (case of three cavities), the higher the temperature detected due to the lower thermal conductivity, and the worse heat transfer capacity of the resulting structure.

### 3.4. Thermal Conductivity of Nanocomposites as Function of Air Cavities Radiuses: Experimental, Theoretical and Simulation Results

Figure 14 reports the experimental results regarding the thermal conductivity of samples based on pure and filled PLA (in (a) and (b), respectively) containing equal volumes of cylindrical and truncated-conical (with two different radiuses) air cavities. The aim of this investigation is to highlight the influence of the radiuses’ sizes on the thermal conductivity. For practical reasons, such dimensions are newly recalled in the same graphs. Before commenting on the results, it should be noted that in these cases, unlike with the previous ones, the cross-sectional area of the air cavities varies along the thickness (obviously except for the cylinder configuration, which can be considered as the particularization of a cone trunk with two numerically equal radiuses). Regardless of the presence or absence of graphene nanoparticles within the composite, the thermal conductivity increases as the product of the two radiuses becomes longer. Therefore, the air cavity with cylindrical shape shows slightly higher conductivity values (0.152 and 0.304 W/mK for pure PLA and PLA + 6 wt% GNPs, respectively) than those measured in the case of truncated-conical-shaped air cavities (0.145 and 0.150 W/mK or 0.290 and 0.299 W/mK, depending on the case). As evident from the analysis of Figure 14c,d, the thermal conductivity dependence on this dimensional product is of a linear type given the values of the regression coefficients (R^2^) close to 1 for the fitting curves of the experimental data.

The results can, once again, be interpreted in light of thermal circuity theory, as schematized in Figure 15a,b.

In this case, reference is made to the calculation of a thermal resistance of a truncated-conical-shaped air cavity with radiuses a and b (a < b), height d and thermal resistivity λ_air_ (see Figure 15c).

The resistance and, therefore, the conductance of the nanocomposite structure remain unchanged, while it is different for the air cavity as the area varies with the radius, which depends on the z coordinates and occurs in a different way.

Furthermore, passing from the greater base to the smaller base, the following expression can be reported for the radius *r*(*z*):(17)$$r\left(z\right)=b-\frac{b-a}{d}\cdot z\;\;\;$$

Otherwise, passing from the minor base to the major base, the radius *r*(*z*) can be expressed as follows:(18)$$r\left(z\right)=a+\frac{b-a}{d}\cdot z$$

Consequently, the thermal conductance of the cavity ($Kth_{air}$) and, therefore, its thermal resistance can be expressed as follows:(19)$$Kth_{air}=\frac{\lambda_{i}\cdot A_{i}}{l_{i}}=\frac{\lambda_{i}\cdot \pi r{\left(z\right)}^{2}}{l_{i}}=\frac{1}{Rth_{i}}$$

These values change according to the radius and in a different way according to the verse in which the air bubble is crossed by the thermal flow. 

Naturally, the average resistance of the cavity remains the same regardless of the verse of the heat flows, as can be observed from the demonstration given below in the case it crosses the cavity from the larger base towards the smaller base.

Once the radius *r*(*z*) has been defined, the elemental resistance *dR* is evaluated as follows:(20)$$dR=\frac{1}{\lambda_{air}}\cdot \frac{dz}{\pi r^{2}}=\frac{1}{\lambda_{air}}\cdot \frac{dz}{\pi {\left[b-\frac{b-a}{d}\cdot z\right]}^{2}}$$

However, the total thermal resistance *R_th_* is obtained via the integration of Equation (20) along the entire height d, in agreement with the following relation:(21)$$R_{th}=\int_{0}^{d}dR=\int_{0}^{d}\;\frac{1}{\lambda_{air}}\cdot \frac{dz}{\pi {\left[b-\frac{b-a}{d}\cdot z\right]}^{2}}=\frac{1}{\lambda_{air}}\cdot \frac{d}{\pi \cdot a\cdot b}$$

At the time that the two radiuses *a* and *b* are equal (*a* = *b* = *r*), Equation (21) degenerates to the case of a cylindrical cavity with thermal resistance *R_th_* equal to the following:(22)$$R_{th}=\frac{1}{\lambda_{air}}\cdot \frac{d}{\pi \cdot a\cdot b}=\frac{1}{\lambda_{air}}\cdot \frac{d}{\pi \cdot r^{2}}$$

The same result is obtained when the cavity is crossed from the minor base to the major base.

The thermal resistance decreases punctually as the radius increases from *a* to *b*; in contrast, it increases punctually when the radius decreases from *b* to *a*.

Naturally, the thermal conductance increases when the resistance decreases and vice versa decreases when the resistance increases, being inverse to each other.

The mean resistance remains the same and changes only when the cavities radiuses a and b and, therefore, their product *a*⋅*b* vary.

Alongside these theoretical considerations, the thermal behaviors of these additional 3D printed discs with designed ad hoc air cavities are also numerically investigated. Figure 16a–c shows (on the left side) the temperature profile (average values in K) for the upper and lower surfaces of the three specimens considered here.

The temperature difference between the two opposite faces (ΔT = 6.92 K, 7.56 K and 8.30 K, respectively) is assumed as an index to uncover, in accordance with Fourier’s thermal law, discrepancies in the thermal conductivity values of the designed structures. The numerical investigation confirms the experimental data regarding the best thermal performance of the sample with cylindrically shaped air cavity compared to the truncated-cone ones.

Moreover, on the right side of Figure 16, the 3D sectional temperature views (at time *t* = 150 s) of the considered specimens are reported with the aim of visually inspecting the heat distribution within the solids. 

A greater local heating is noted especially in the case of the truncated-cone cavity with a larger base (Figure 16c) since it hinders the thermal flow and consequently raises the surrounding temperature.

### 3.5. Thermal Conductivity of Nanocomposites as Function of Air Cavities Orientation: Experimental, Theoretical and Simulation Results

To conclude the thermal analysis conducted in the present study, this subsection takes into consideration, with reference to the truncated-conical-shaped air cavity, the effect of its orientation on the thermal conductivity of the resulting structure. Figure 17 shows the experimental results (for pure PLA and PLA + 6 wt% GNPs in (a) and (b), respectively) concerning the thermal conductivity of the specimens with truncated-conical air cavities when rotated 180° in order to reverse the orientation of the same cavity. It is surprising to note that this overturning of the air cavity affects the thermal conductivity. Furthermore, the same sample exhibits a higher conductivity value when the air cavity is placed with the larger radius on the top or similarly with the lower radius downwards at the face from which the sample is heated.

For specimens realized with pure PLA, the thermal conductivity changes from 0.145 to 0.151 W/mK and from 0.150 to 0.155 W/mK in the two investigated cases, whereas for GNPs-based composites, the variation is from 0.290 to 0.296 W/mK and from 0.299 to 0.305 W/mK, respectively.

The interpretation of this result appears to be slightly more complicated. In a first analysis, reversing the orientation of the cavity, it soon becomes clear that the variation in the cross-section area along the thickness changes in the two cases: in the increasing direction of the z-axis, in one case, it decreases; in the other, it increases. Most likely, this affects the thermal diffusion within the medium in some way as is more evident from the graphical analysis of Figure 18a–d, which presents the 3D cross section temperature views for the different combinations.

When the larger base of the air cavity is turned downwards in correspondence with the face of the sample on which the thermal source is applied, an evident local overheating is detected. The heat in diffusing upwards inevitably encounters a wider barrier (air with lower conductivity with respect to that of material) which hinders its propagation towards the opposite upper face of the specimens.

By exploiting the potential of the simulator, it is possible to deepen the topic in order to better understand the thermal diffusion inside the solid in these situations, which is difficult to explore otherwise. From here onwards, the numerical simulations will focus exclusively on the truncated cone air cavity with a large radius of 4.000 mm. This is because its cross-sectional area variation along the thickness is wider than that of the truncated cone with a radius of 3.500 mm.

Figure 19 shows the changes in temperature over time (T_MAX_, T_AVERAGE_ and T_MIN_ evaluated on the entire domain) up to 150 s (top part of Figure) in the case of an air cavity with a larger base down- and up-oriented in (a) and (b), respectively.

Comparing the two graphics, the different thermal behavior immediately stands out: a larger thermal hysteresis appears in the results depicted in Figure 19a with respect to those shown in Figure 19b. Even the peak of the maximum temperature detected appears to be definitely more pronounced in the case of an air cavity with a large base facing down (approximately 325 K against 320 K, to be exact), as well as that of the difference between the two temperature T_MAX_-T_MIN_ reported in the inserts of the same figures (16 K against 14 K). All these observations are in favor of a better thermal behavior of the composite with the air cavities with the larger base facing up, as experimentally observed. Moreover, the same Figure 19 reports (in the respective lower parts) a particular 3D sectional view of the samples that better highlights the thermal distribution previously discussed with reference to the 2D views of previous Figure 18.

By deepening this numerical analysis, the different temperature profiles were observed in correspondence with some suitably selected segments across the entire thickness, as shown in Figure 20a,b, respectively. Furthermore, as shown in the same figure, the first segment identified by the coordinates x = 0, y = 0 and z = 0 ÷ 3 corresponds to the axis of symmetry of the sample, whereas the second segment (x = 2.554, y = 0 and z = 0 ÷ 3) was selected because it crosses the center of the oblique side of the air cavity exactly.

The temperatures are recorded at some specific instants of time until thermal equilibrium is reached (150 s) once the heat flux has been suppressed (60 s). The simulation results are shown in Figure 20c–f, depending on the considered case.

It is important to highlight how, regardless of the case and the instant of observation time, the slope of the temperatures varies along the thickness: it increases as soon as the heat crosses the section that includes air with lower thermal conductivity. With reference to Figure 20c,d, it can be observed that the temperature values along the selected segment through the entire thickness (axis of symmetry in these cases) are appreciably higher when the air cavity exposes its major base downwards, especially after the first instants of time, after which the heat spreads with greater inertia due to the presence of air. The same considerations are also valid in the cases reported in Figure 20e,f where the difference, with respect to that analyzed above, lies only in the position of the selected segment and therefore depends on the portion of material or material/air that the heat encounters in its propagation. These additional numerical results also support the experimentally observed results.

Moreover, temperature differences (ΔTi with i = 1,…,6) are evaluated between a reference level (it is assumed at quote z = 0 mm) and some particular cross-sections of the simulated sample, as schematically represented in Figure 21a. Some of them intersect solids exclusively in the material (at z-levels equal to 0.5, 2.5 and 3 mm) and others material/air (at z-levels equal to 1, 1.5 and 2 mm). The numerical results (average values on the surfaces), in the event that the air cavity has the larger base facing downwards or upwards, are reported in Figure 21b,c, respectively.

Comparing these results, it can be observed that, as expected, the first temperature difference (ΔT_1_) is the same in both cases because the heat in the first phase and in the considered sections (z = 0.5 mm) diffuses exclusively in the material, which is common. The subsequent evaluations (ΔTi with i = 2, …, 6), on the other hand, differ from each other because the diffusion of heat encounters material/air sections that differ in composition. The sample with the air cavity with a larger base facing upwards (Figure 21c) always shows significantly lower temperature differences than those in which the cavity has the opposite orientation (Figure 21b). This results in a more uniform distribution of heat inside the solid due to its more efficient heat transfer. Therefore, this further numerical result is also in line with the different values of thermal conductivity experimentally measured.

Finally, to conclude this numerical investigation, Figure 22 shows the results regarding the variation in total internal energy over time (up to 150 s) for the two cases of orientation of the cavity (in (a) and (b), top side) considered in the present study, whereas the corresponding 3D views (at t = 60 s) are presented on the lower parts of the same figure.

Regardless of the type of total internal energy observed (maximum, minimum, average value or difference between maximum and minimum) in the presence of an air cavity with larger base facing upward, the internal energy is clearly less compared to that with opposite orientation. These results reflect the previously estimated temperature distributions in the two samples once again since the energy variation ΔU is linked to them by the following analytical relationship:(23)$$\Delta U=U\left(T_{f}\right)-U\left(T_{i}\right)$$
where *T_f_* and *T_i_* indicate the initial and final temperatures of the solid, respectively. The greater the difference between the two temperatures, the greater the change in energy. The three-dimensional views of some sections of the internal energy allow us to investigate its distribution within the structures; in any case, maximum values are found in correspondence with the air cavity due to the previosuly discussed local overheating.

## 4. Discussion

In recent years, academic and industrial research efforts have been dedicated to the design of novel and advanced composites using nanoscale fillers. The main goals are the reduction in mass with consequent savings in materials and costs, and the improvement of the overall properties, including the mechanical, electrical and thermal ones. Today, thermal-conductive polymer composites are increasingly assessed for heat transport applications as alternatives to classic metals, such as aluminum and copper, which are high-density materials. Moreover, polymer composites present a lot of benefits over conventional materials, which include easy processability, a higher strength-to-weight ratio, high resistance to corrosion, and lower costs. In the present paper, the thermal conductivity of polymer nanocomposites based on pure PLA and filled with 6 wt% of graphene nanoplatelets (GNPs) has been experimentally, theoretically and numerically investigated.

Some disc-shaped specimens have been prepared via additive manufacturing (AM, also known as 3D printing) based on fused deposition modelling (FDM), which is one of the most widely used technologies as a result of its simplicity. Briefly, a thermoplastic filament is melted and then extruded by a nozzle to build, on a bed-printing and layer-by-layer process, the object according to a computer-aided design (CAD) model. The recent progress in such fabrication processes, in terms of technology and materials that can be used as feedstocks and dimensional accuracy of the 3D printed parts, has paved the way for even a wider range of flexibilities in design and prototyping [44]. Benefiting from such potential, the test specimens considered in the present study were provided with some ad hoc designed air cavities to investigate the influence of their geometry and orientation on the overall thermal properties of the resulting structures. 

A preliminary morphological analysis has revealed a good interaction filler/matrix, as well as a particular stacked arrangement of the graphene nanoparticles, which favors the thermal transport due to the reduction in the interfacial and interparticle thermal resistance. By introducing an amount of 6 wt% of GNPs, the experimental results show an improvement of the thermal conductivity of approximately 101% compared to that of neat PLA.

Similar interesting results are reported in the literature with reference to other thermoplastic polymers, such as polycarbonate [45] or thermosetting ones (including epoxy resins [46,47]). The roles of filler introduction, and of the air cavity geometries and their orientation have been systematically investigated first in an experimental manner, and then interpreted with theoretical and simulation studies. Furthermore, a laser flash analysis (LFA), based on non-contact measurements, is used to measure the thermal properties of the designed structures since it is widely recognized as a fast, versatile and precise absolute method for a variety of different materials and temperatures of interest. For a long time, the thermal conductivity of polymers has been investigated within the academic and industrial communities. The existing theories can be successfully applied to predict the thermal conductivity of the polymers reinforced with conventional-sized microparticles [48]. What has proven to be more complicated is the ability to estimate the thermal conductivity behavior of the recent class of composites based on a filler of nanometer size since it depends on a great variety of affecting parameters, such as the interfacial polarization, filler shape and aspect ratio, and its dispersion state, among others. Kochetov et al. have proposed a model specifically designed for the thermal conductivity of the polymer nanocomposites which simultaneously accounts for the physical properties of the polymer matrix, the nanofiller and their interfacial interaction [49]. Lambin et al. have presented numerical calculations and an efficient homogenization theory to evaluate the thermal properties of 3D printed multifunctional structures with holes made of composite polymers filled with nanocarbon particles [43]. Vega-Flick et al. have presented a review of recent studies of thermal transport in nano-structured materials including methodologies and analysis of measurements errors [50]. In the present study, to interpret the experimental results, a combination of the theories mentioned above with the basic notions of thermal circuits has been used. There is indeed an electrical analogy with conduction heat transfer that can be applied to problem solving [51]. Today, computational modeling is a well-established and consolidated activity in the field of material science for exploring, selecting, and designing purposes [52,53]. Therefore, in the current work, multiphysics simulations based on a finite element method (FEM) have been performed in order to support the experimental results and better understand the complex thermal phenomena that are manifested in such complex structures.

It has been observed that with the same volume of air cavities and when the latter have a constant cross-sectional area along the thickness, the geometry does not influence the thermal conductivity. Instead, it is conditioned and depends inversely on the products of the radiuses of the air cavities in the case in which the area of their cross section is variable. Finally, an important role is also played by the orientation of the cavity in the case in which they are truncated cone-shaped. Further thermal aspects and more complex air cavities will be considered in a future paper to investigate the potential use of lightweight and thermally conductive polymers in heat transfer applications.

## 5. Conclusions

This paper principally addressed the investigation of the thermal behavior of composites based on pure poly-lactic acid (PLA) and filled with a suitable amount of graphene nanoplatelets (GNPs) produced via additive manufacturing (AM) of type fused deposition modeling (FDM). Filaments (of our own production) of pure PLA and filled with 6 wt% of GNPs have been used as feedstocks. Furthermore, benefiting of the extraordinary versatility of design of the 3D printing technology, some disc-shaped specimens including ad hoc designed air cavities have been prepared and then experimentally, theoretically and numerically investigated. The aim of this paper has been to investigate the influence of the incorporation of nanosized fillers, as well as the influence of the different air cavity geometries and their orientations on the overall thermal conductivity of the resulting structures. Although the importance of theoretical studies is obvious, in light of the fundamental support of the modelling activity to the experimental results, computational approaches are encouraged in material science to discover new advanced materials or to deepen the knowledge of the existing ones in order to proceed with their optimization. As the topic is of great interest in the field of polymers, further studies will be devoted to investigating increasingly complex and lightweight structures without the loss of overall performance.

## Figures and Tables

**Figure 1 nanomaterials-13-01863-f001:**
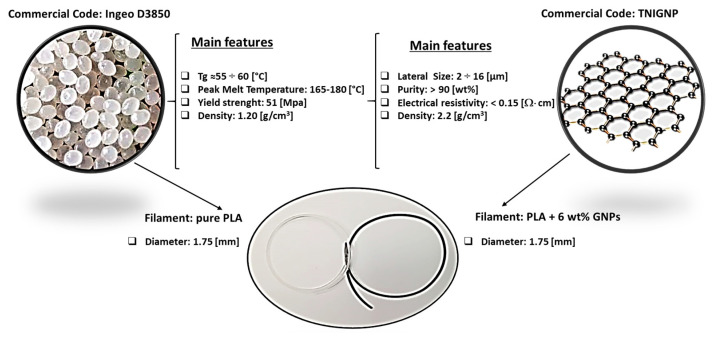
Main physical properties of the polymer, graphene nanoplates and manufactured filaments adopted in the present work.

**Figure 2 nanomaterials-13-01863-f002:**
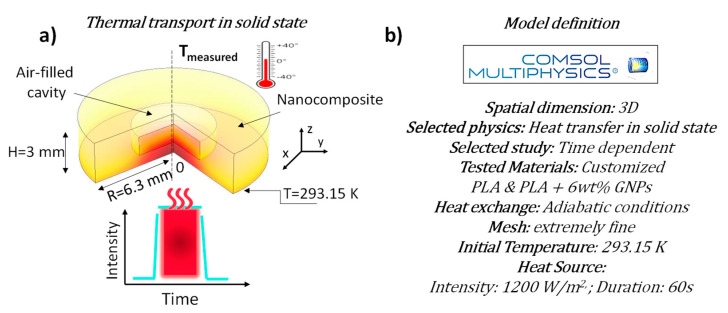
(**a**) Schematic representation of the simulated case study. (**b**) Key model definitions adopted for the multiphysics simulations.

**Figure 3 nanomaterials-13-01863-f003:**
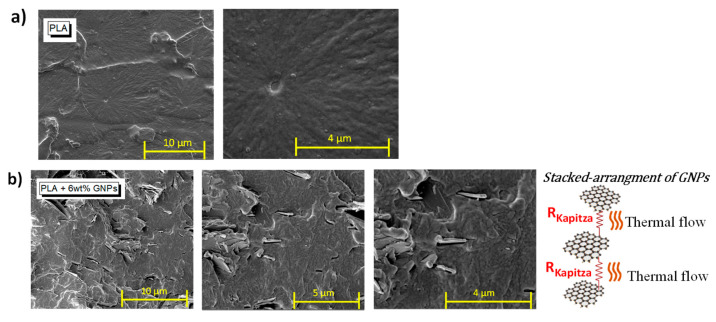
Scanning electron microscopy (SEM) analysis of pure PLA in (**a**) and nanocomposites filled with 6 wt% of GNPs in (**b**), respectively. A schematic representation of the stacked arrangement of the GNPs is also reported in (**b**). A micrometer scale is due to the large lateral size (mean diameter) of 2–16 µm of the adopted GNPs.

**Figure 4 nanomaterials-13-01863-f004:**
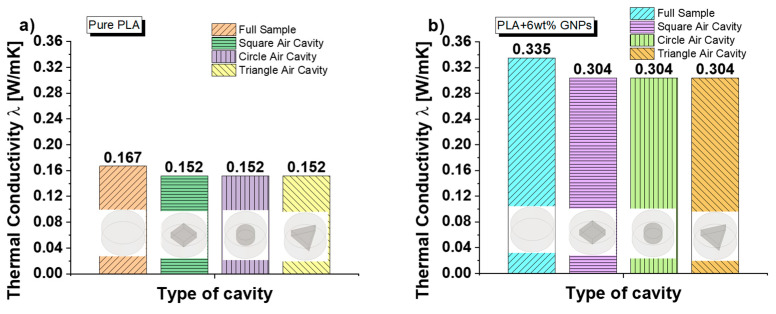
Experimental results on the thermal conductivity for full samples and those including different shapes of air cavity for pure PLA in (**a**) and nanocomposites filled with 6 wt% of GNPs in (**b**), respectively.

**Figure 5 nanomaterials-13-01863-f005:**
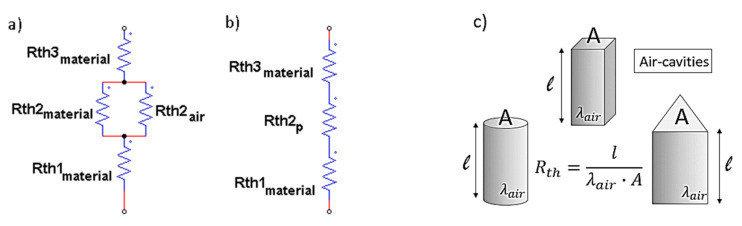
Thermal resistance circuits for the evaluation of the parallel and series conductivity in (**a**) and in (**b**); geometric parameters and intrinsic properties associated with the different air cavities in (**c**).

**Figure 6 nanomaterials-13-01863-f006:**
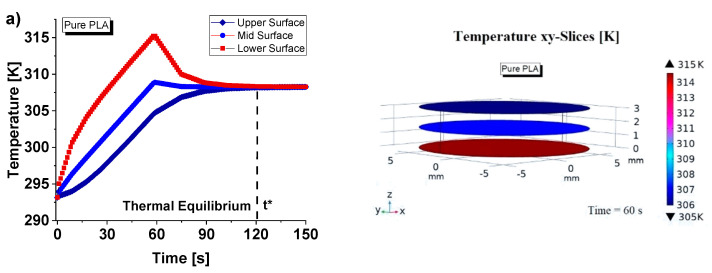
Comparison of the temperature profiles (average values) recorded on the upper, mid and lower surfaces vs. time for composite based on pure PLA and PLA + 6 wt% GNPs in (**a**) and (**b**), respectively. In particular, bidimensional graphics are reported on the left part, whereas a 3D view of the corresponding xy-slices (at time *t* = 60 s) is reported on the right part.

**Figure 7 nanomaterials-13-01863-f007:**
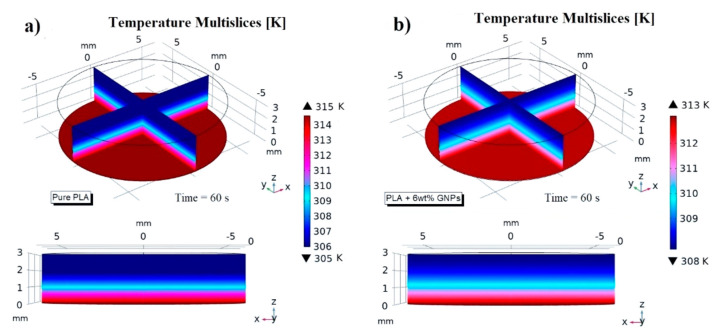
Temperature multislice views evaluated at *t* = 60 s relatively to pure PLA, and PLA + 6 wt% GNPs in (**a**) and (**b**), respectively.

**Figure 8 nanomaterials-13-01863-f008:**
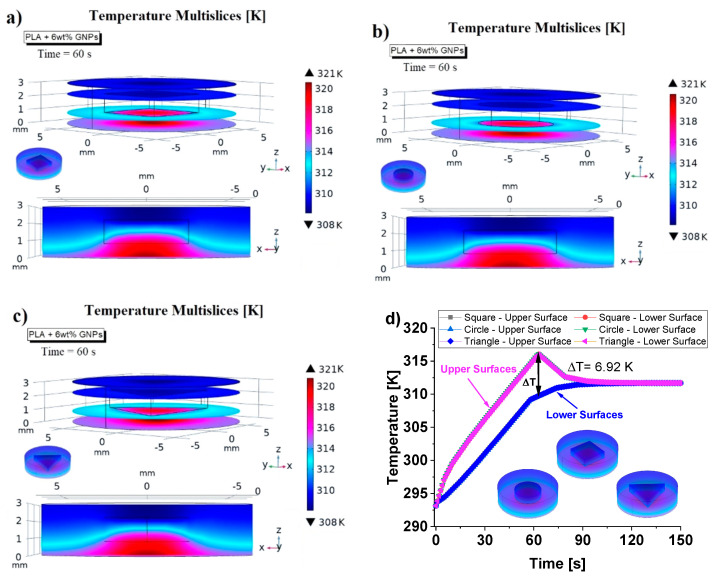
Temperature multislice views (at t = 60 s) of the samples including a variable shape of air cavity such as square in (**a**), circle in (**b**) and triangle in (**c**), respectively. The dynamic temperature evolutions over the time (up to 150 s) for the three different cases are reported in (**d**).

**Figure 9 nanomaterials-13-01863-f009:**
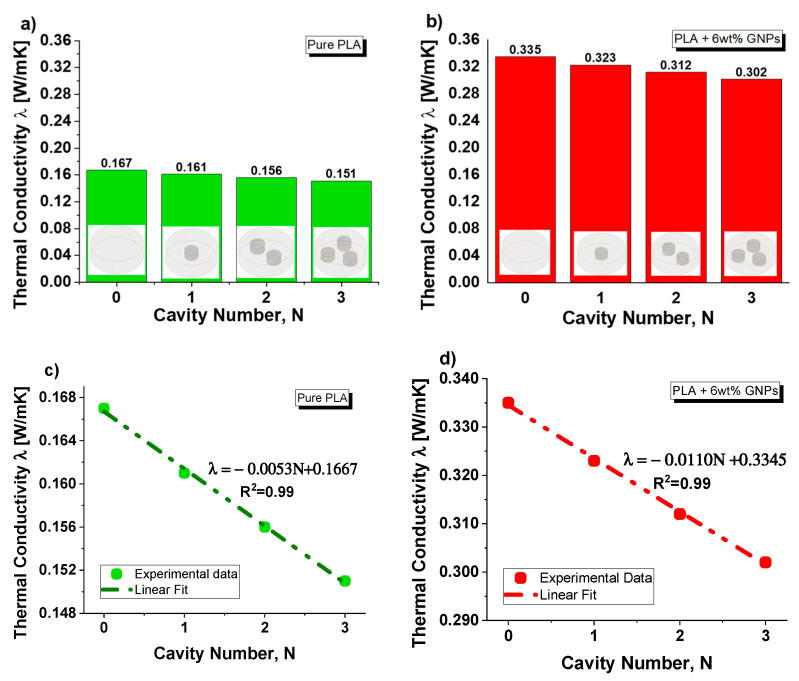
(**a**) Thermal conductivity as function of the cavity number for composites based on pure PLA in (**a**) and PLA + 6 wt% GNPs in (**b**). A linear fit of the experimental data is reported in (**c**) and (**d**) for pure and filled PLA, respectively.

**Figure 10 nanomaterials-13-01863-f010:**
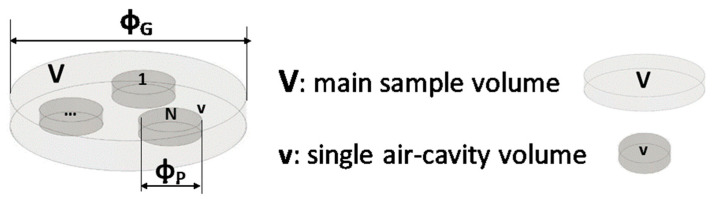
Schematic representation for the specimens filled with multiple air cavities.

**Figure 11 nanomaterials-13-01863-f011:**
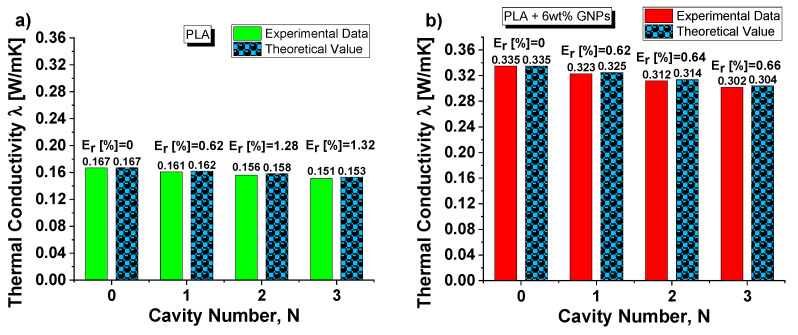
(**a**) Comparison between the experimental and theoretical results, together with the relative error (in percentage) that is committed with the theoretical estimation, about the thermal conductivity as function of the cavity number for composites based on pure PLA in (**a**) and PLA + 6 wt% in (**b**).

**Figure 12 nanomaterials-13-01863-f012:**
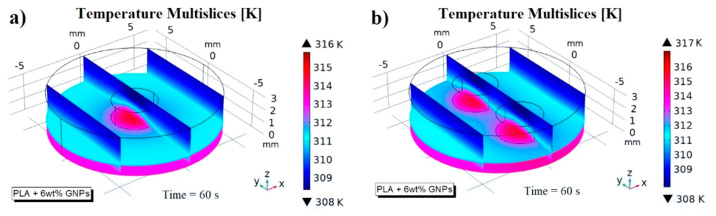
Temperature multislice views for composites including different number of cavities (1, 2 and 3) in (**a**), (**b**) and (**c**), respectively.

**Figure 13 nanomaterials-13-01863-f013:**
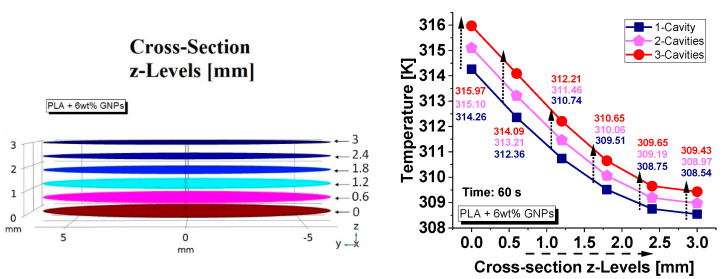
Cross sections along the thickness of the sample (**left side**) on which to detect the average temperature values (**right side**) at time t = 60 s for samples with a variable number of air cavities.

**Figure 14 nanomaterials-13-01863-f014:**
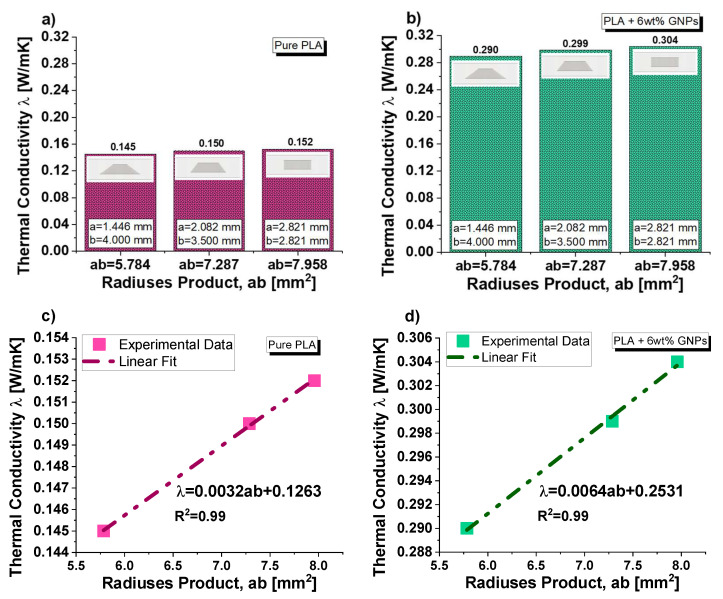
(**a**) Thermal conductivity as a function of the radiuses product for composites based on pure PLA in (**a**) and PLA+6 wt% in (**b**). A linear fit of the experimental data is reported in (**c**) and (**d**) for pure and filled PLA, respectively.

**Figure 15 nanomaterials-13-01863-f015:**
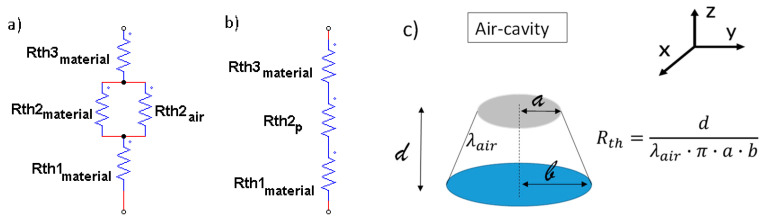
Thermal resistance in circuit in (**a**) and truncated-conical shape air cavity with geometrical features in (**b**); geometric parameters and intrinsic properties associated with the truncated-conical air cavities in (**c**).

**Figure 16 nanomaterials-13-01863-f016:**
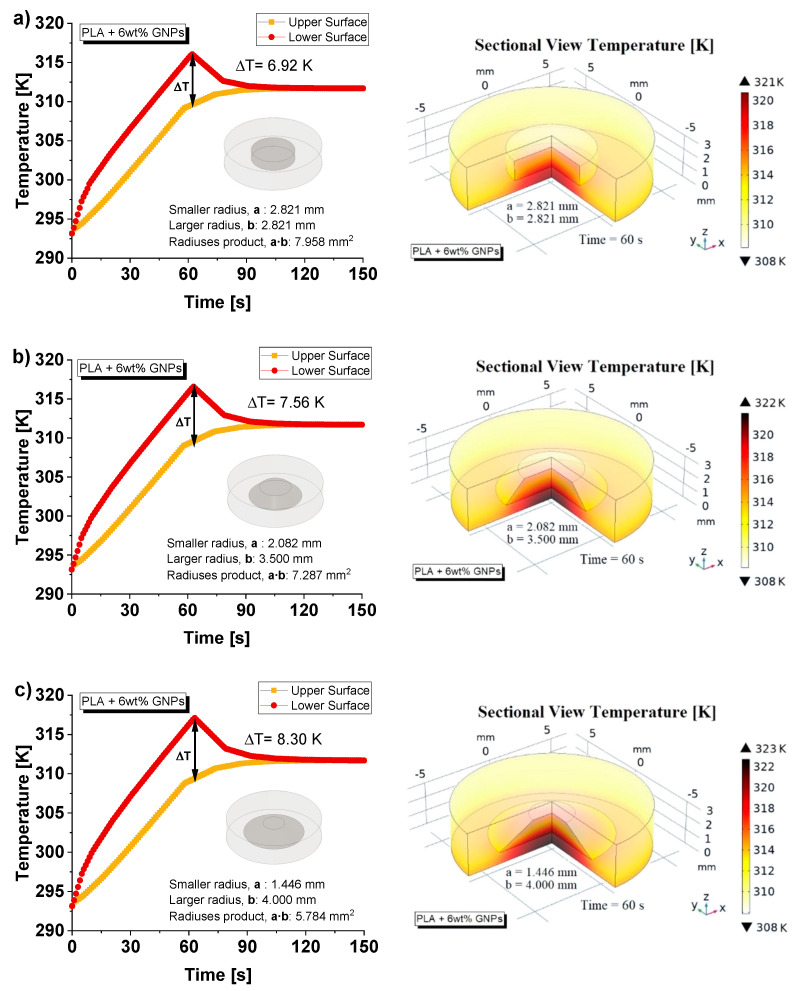
(**a**) Numerical analysis of the temperature evolution of the upper and lower surfaces over the time (up to 150 s) for the 3D printed discs with designed ad hoc air cavities. The case of cylindrical-shaped and truncated-conical cavities with different radiuses are considered on the left side of (**a**), (**b**) and (**c**), respectively. On the respective right-hand parts, sectional views of temperature are shown to visually explore the different heat distribution within the solids.

**Figure 17 nanomaterials-13-01863-f017:**
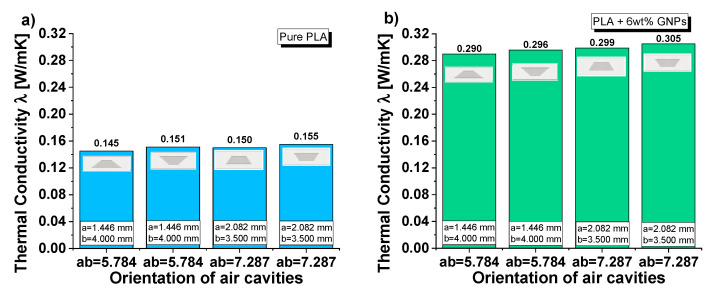
Experimental results on the thermal conductivity as a function of the air cavity orientation for composites based on pure PLA in (**a**) and GNPs-based composites in (**b**).

**Figure 18 nanomaterials-13-01863-f018:**
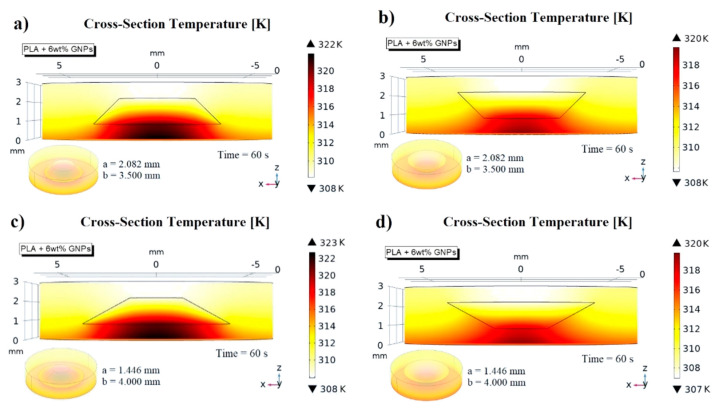
(**a**) Cross-section temperature views for the possible orientation of the air cavity: cases for truncated-cone with a large radius of 3.500 mm in (**a**,**b**) and for truncated cone with a large radius of 4.000 mm in (**c**) and (**d**), respectively.

**Figure 19 nanomaterials-13-01863-f019:**
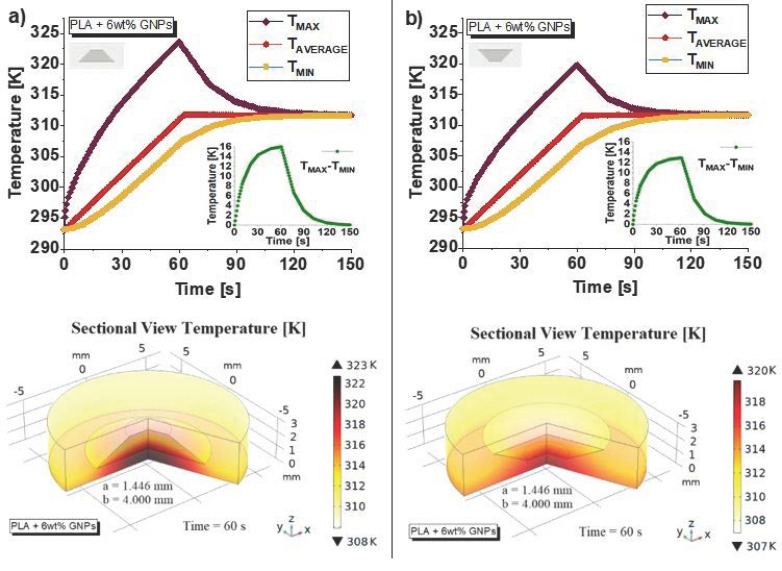
Temperature evolutions (T_MAX_, T_AVERAGE_ and T_MIN,_ evaluated on the entire domain) over time up to 150 s (top part of Figure) in the case of an air cavity with a larger base down- and up-oriented in (**a**) and (**b**), respectively. The respective inserts show the difference between the two temperatures TMAX–TMIN. In the lower parts of the figure, the 3D sectional views of the overall temperature of the two solids at time t = 60 s are reported.

**Figure 20 nanomaterials-13-01863-f020:**
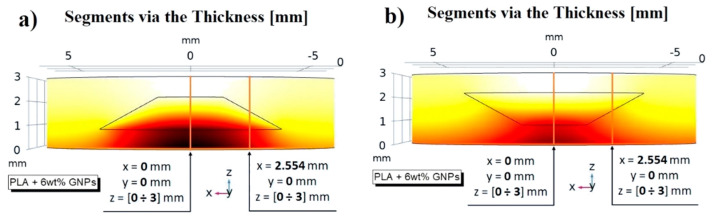
Selected segments through the entire thickness on which to evaluate the temperature profiles for both orientations of the air cavity in (**a**) and (**b**), respectively. The recorded temperatures on these segments are reported in (**c**–**f**), depending on the case.

**Figure 21 nanomaterials-13-01863-f021:**
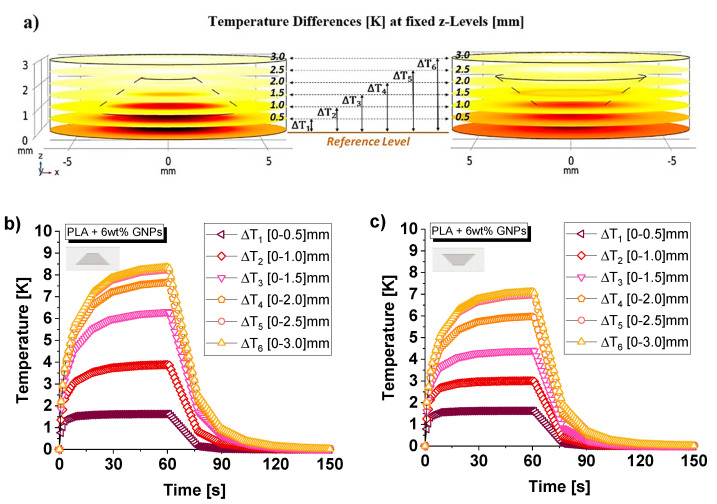
Some selected cross-sections for the evaluation of the temperature differences with respect to the reference level in (**a**). Numerical results for both possibilities of orientations of the air cavity: with the larger base facing down and up in (**b**) and (**c**), respectively.

**Figure 22 nanomaterials-13-01863-f022:**
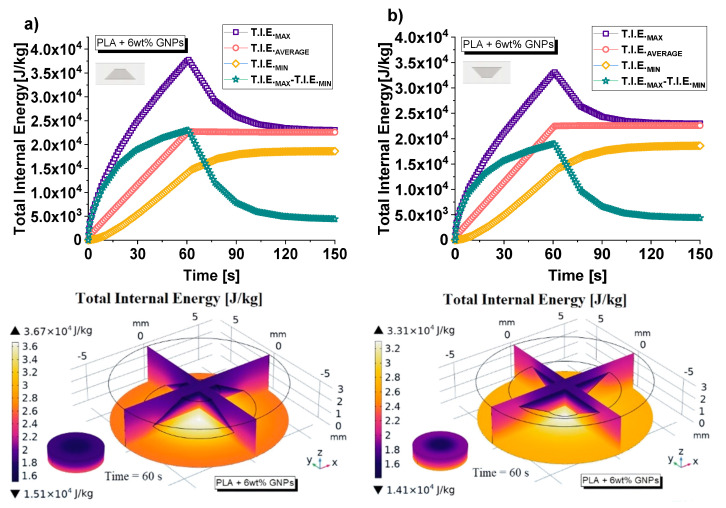
Change in total internal energy (in all domain) over time (up to 150 s) for both possibilities of orientation of the air cavity: with the larger base facing down and up in (**a**) and (**b**), upper sides, respectively. In the corresponding lower parts of the figure, 3D views of the total internal energy at time t = 60 s.

**Table 1 nanomaterials-13-01863-t001:** List of samples designed for evaluating the influence of geometry of air cavity.

Sample	Air Cavity Geometry	Data Air Cavity	Mathematical Formalism
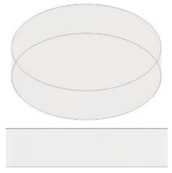	Full sample	//	//
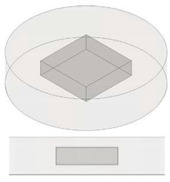	Square cavity 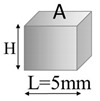	**Fixed Side**, L: 5 [mm];**Fixed Height**, H: 1.4 [mm];**Calculated Area, A:**A: 25 [mm^2^]**Calculated Volume, V:**V = 35 [mm^3^]	Area as function of L A=L·L
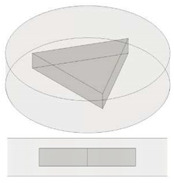	Equilateral Triangular cavity 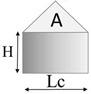	**Fixed Area,** A-Area: 25 [mm^2^]**Fixed Height,** H: 1.4 [mm];**Fixed Volume** V: 35 [mm^3^]**Calculated Side, L_C_:**Lc = 7.598 [mm];	Area as function of Lc A=34·LC2
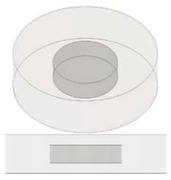	Circular cavity 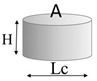	**Fixed Area,** A-Area:25 [mm^2^]**Fixed Height,** H: 1.4 [mm];**Fixed Volume** V: 35 [mm^3^]**Calculated Diameter, L_C_:**Lc = 5.642 [mm];	Area as function of Lc A=π·LC24

**Table 2 nanomaterials-13-01863-t002:** List of samples designed for evaluating the influence of volume of air cavity.

Sample	Air Cavity Geometry	Data Air Cavity	Mathematical Formalism
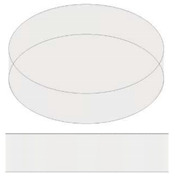	Full sample//	//	//
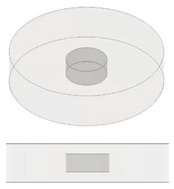	1 Circular cavity 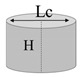	**Fixed diameter, L_C_:**3.385 [mm] **Fixed Height,** H: 1.4 [mm];**Calculated Area, A:**A: 9.005 [mm^2^]**Calculated Volume, V:**V = 12.606 [mm^3^]	Area as function of Lc A=π·LC24
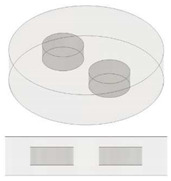	2 Circular cavities 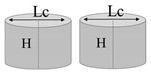	**Fixed diameter, L_C_:**3.385 [mm] **Fixed Height,** H: 1.4 [mm];**Calculated Area, A:**A = 18.009 [mm^2^]**Calculated Volume, V:**V = 25.213 [mm^3^]	Area as function of Lc Atot=2∗π·LC24
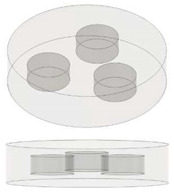	3 Circular cavities 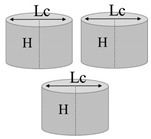	**Fixed diameter, L_C_:**3.385 [mm] **Fixed Height,** H: 1.4 [mm];**Calculated Area, A:**A = 27.014 [mm^2^]**Calculated Volume, V:**V = 37. 819 [mm^3^]	Area as function of Lc A=3∗π·LC24

**Table 3 nanomaterials-13-01863-t003:** List of samples designed for evaluating the influence of the radiused of the truncated cone air cavities and their orientations.

Sample	Air Cavity Geometry	Data Air Cavity	Mathematical Formalism
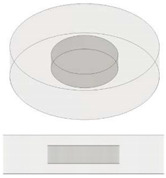	1 Cylindrical cavity 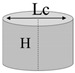	**Fixed Volume:** 35 [mm^3^]**Fixed High,** H: 1.4 [mm];**Calculated A, Area:** 25 [mm^2^]**Calculated radius, a:**A = 2.821 [mm];	Area as function of Lc A=π·a 24
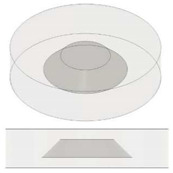	1 truncated cone cavity 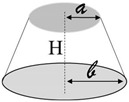	**Fixed Volume, V:** 35 [mm^3^]**Fixed High, H:** 1.4 [mm];**Fixed larger radius, b:** 3.500 [mm]**Calculated minor radius, a:**2.082 [mm];	Volume as function ofradius a,b V=13·π·a2+a·b+b2·H
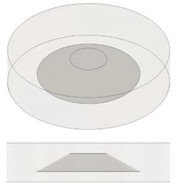	1 truncated cone cavity 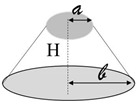	**Fixed Volume, V:** 35 [mm^3^]**Fixed High, H:** 1.4 [mm];**Fixed larger radius, b:** 4.000 [mm]**Calculated minor radius, a:**1.446 [mm];	Volume as function ofradius a,b V=13·π·a2+a·b+b2·H

**Table 4 nanomaterials-13-01863-t004:** Initial and boundary conditions for solving the thermal energy equation.

Initial (I.C.) and Boundary (B.C.) Conditions		Equations	Validity
I. C.	t = 0	T = Room Temperature (T_0_)	∀x,∀y,∀z
B. C.	Lower Surface	$-\lambda \frac{\partial T}{\partial z}=q_{0}$	(∀x,∀y,t>0)
B. C.	Upper Surface	$-\lambda \frac{\partial T}{\partial z}=0$	(∀x,∀y,t>0)
B. C.	Lateral Surfaces	$-\lambda \frac{\partial T}{\partial y}=0;-\lambda \frac{\partial T}{\partial x}=0$	(∀z,t>0)

## Data Availability

Data are contained within the article.

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
