# Peer review of "Experimental, Theoretical and Numerical Studies on Thermal Properties of Lightweight 3D Printed Graphene-Based Discs with Designed Ad Hoc Air Cavities"

_nanomaterials, 2023, doi:10.3390/nano13121863_

Round 1

Reviewer 1 Report

The paper entitled “Experimental, Theoretical and Numerical Studies on Thermal Properties of Lightweight 3D-printed Graphene-based Discs with Designed Ad-hoc Air-cavities” contains valuable experimental and theoretical results. The topic is interesting and has huge importance. Some minor typing mistakes are present, and a lot of detailed things which do not belong directly to the scientific result of the paper should be shifted to the Electronic supplementary materials.

The evaluation of results is correct, and the building of the paper is good.

Some examples:

1., PLA abbreviation is not resolved in the abstract (line 17).

2. Excess hyphen (line 89)   

3. Fig.2, Table 1 contains data without scientific importance and should be shifted to ESI.

4. The theoretical background of the measurements, which have been published elsewhere should be cited only, and the detailed description should be shifted to ESI lines 177-184.

5. The simulation conditions (from line 242) should be discussed in the results and discussion (or partly to ESI).

6. Line 418, typing mistake (experimental)

7. Fig.13 and Table 6 contain the same data. One among them should be shifted to ESI.

Thus I can suggest publishing the paper after minor revision.

Author Response

First of all, thank you for the time and efforts spent in evaluating our paper. 

Reviewer 2 Report

In this paper, the thermal behavior of composite materials based on pure polylactic acid (PLA) and filled with appropriate amount of graphene nanosheets (GNPs) prepared by additive manufacturing (AM) melt deposition model (FDM) was investigated. The paper has sufficient data and standardized writing. I suggest to publish it after minor revision. The modification suggestions are as follows:

1. Please arrange the SEM  in Figure 5 neatly. And remove the instrument information in the SEM image.

2, The inset in Figure 5 should be properly enlarged and the font should be improve.

3. Add the display of microscopic morphology at the nanoscale.

4. Add the XRD pattern of the graphene nanosheets used.

5. Optimize the presentation of Figure 12, so that readers can better understand the content of the figure.

6. Combine the results and discussion, and present the results while having a discussion.

7. Appropriately increase the reading and citations of top journals.

Author Response

(The authors gave the same response as above.)
